# Estimating the effects of temperature on transmission of the human malaria parasite, *Plasmodium falciparum*

Eunho Suh [1,7] ✉, Isaac J. Stopard [2,7], Ben Lambert [3], Jessica L. Waite[1,5], Nina L. Dennington[1], Thomas S. Churcher [2] & Matthew B. Thomas[1,4,6]

Despite concern that climate change could increase the human risk to malaria in certain areas, the temperature dependency of malaria transmission is poorly characterized. Here, we use a mechanistic model fitted to experimental data to describe how *Plasmodium falciparum* infection of the African malaria vector, *Anopheles gambiae*, is modulated by temperature, including its influences on parasite establishment, conversion efficiency through parasite developmental stages, parasite development rate, and overall vector competence. We use these data, together with estimates of the survival of infected blood-fed mosquitoes, to explore the theoretical influence of temperature on transmission in four locations in Kenya, considering recent conditions and future climate change. Results provide insights into factors limiting transmission in cooler environments and indicate that increases in malaria transmission due to climate warming in areas like the Kenyan Highlands, might be less than previously predicted.

Malaria is a leading cause of morbidity and mortality in many tropical and sub-tropical regions[1]. The transmission of malaria parasites is inextricably linked to the biology of the mosquito vector. Classic models that estimate the Basic Reproduction Number ($R_0$) or Vectorial Capacity[2–4] indicate that traits such as mosquito life expectancy, vector competence, per-mosquito biting rate, and the Extrinsic Incubation Period (EIP; the duration of sporogony) are important determinants of malaria transmission. Despite their epidemiological importance, these traits are poorly characterised for most malaria vectors. A limited understanding of these traits and their interaction with environmental variables, such as temperature, constrains our ability to understand current patterns of transmission or future responses to climate change.

The EIP describes the time it takes for a mosquito to become infectious following an initial infected blood meal. This process involves parasites infecting the midgut, undergoing obligate sexual and asexual reproduction via multiple life stage transitions and then migrating to the mosquito salivary glands where they may be secreted into saliva[5,6]. Mosquitoes are ectothermic and although the EIP can be influenced by different biotic and abiotic factors[7], the single biggest known determinant is ambient temperature[8–12]. For many decades, the temperature dependence of the EIP for *Plasmodium falciparum*, the most important human malaria-causing parasite, has been characterised using a single degree-day model[7,13]. This model assumes there is a fixed number of degree-days ($D$), or heat units, that need to be accumulated for the completion of sporogony such that,

$$EIP_D = \frac{D}{T - T_{min}},\qquad(1)$$

where $T$ is the mean daily temperature and $T_{min}$ is the lower temperature threshold for parasite development. For *P. falciparum*, $D$ was

[1]Center for Infectious Disease Dynamics, Department of Entomology, The Pennsylvania State University, University Park, PA, USA. [2]MRC Centre for Global Infectious Disease Analysis, School of Public Health, Faculty of Medicine, Imperial College London, London, UK. [3]Department of Statistics, University of Oxford, Oxford, UK. [4]Department of Biology, University of York, York, UK. [5]Present address: Research Development, University of Vermont, Burlington, VT, USA. [6]Present address: Invasion Science Research Institute and Department of Entomology and Nematology, University of Florida, Gainesville, FL, USA. [7]These authors contributed equally: Eunho Suh, Isaac J. Stopard. ✉e-mail: eus57@psu.edu

estimated over 60 years ago to be 111 with $T_{min}$ equal to 16 °C[7,13]. Existing climate sensitive malaria models that account for the effects of temperature on the EIP when predicting malaria endemicity or transmission intensity commonly rely on this model[14–22]. Despite its extensive application, the established degree-day model remains poorly validated, and several potential limitations have been identified[7,14–22]. First, the model was parameterised based on a single minimally replicated experiment[23], with the EIP measured as the first day at which an individual mosquito was observed to have parasites in the salivary glands[7,13]. Recent evidence, however, indicates variation in the EIP between mosquitoes exposed to the same environment[24,25]. Second, there were no empirical measures of the EIP below 19-20 °C, meaning the degree-day model estimates at cooler temperatures are based on extrapolation. However, exploratory experiments measuring EIP below 20 °C indicate this extrapolation may overestimate the EIP[25]. Finally, the original experiment was conducted with *Anopheles maculipennis*, a mosquito species complex responsible for malaria transmission in Eurasia[7]. The EIP is known to vary between *Plasmodium* species[13,26], but it is unclear whether it varies between different mosquito species[7,25] and there has never been an EIP model developed for any dominant vector species in Africa.

Our understanding of the effects of environmental temperature on other important traits, such as the vector competence (the proportion of mosquitoes successfully develop infectious parasites in the salivary glands following an initial infectious blood meal) and mosquito longevity are similarly incomplete. A limited number of studies indicate that parasite establishment in the mosquito is sensitive to temperature[27,28]. There is, however, insufficient data on subsequent *P. falciparum* development in *Anopheles gambiae*, and mechanistic modelling studies examining the influence of climate on malaria transmission tend to either assume the vector competence is temperature invariant, or utilise data from infection experiments using inappropriate mosquito-parasite pairings such as species of North American vector infected with *P. vivax*[9,22,29–31]. Similarly, data on the effects of temperature on the survival of key malaria vectors are also scant, particularly in the field[32].

Here, we generate a comprehensive set of experimental infection data for *P. falciparum* in the dominant African malaria vector, *A. gambiae*, across a range of temperatures. Mosquitoes are dissected to determine the presence of different parasite stages, so it is not possible to observe the parasite population dynamics within single mosquitoes. To estimate temperature-dependent changes in the EIP distribution, vector competence and parasite load among the mosquito population, we fit a mechanistic model[33] that simulates the temporal dynamics of sporogony. We use these estimates, together with data on temperature-dependent mosquito mortality rate, to parameterise a model of vectorial capacity and use this to investigate the effects of temperature on current and future transmission in four locations in Kenya.

## Results

### Experimental infection of *P. falciparum* in *A. gambiae* mosquitoes at different temperatures

We began with a pilot study to refine sample sizes and to better understand the relationship between gametocytemia (percent gametocyte infection in red blood cells) and infection dynamics at different temperatures. *A. gambiae* mosquitoes (G3) were fed infectious blood meals with *P. falciparum* (NF54) gametocytemia of 0.024% within a standard membrane feeding assay and then maintained at constant temperatures of 17, 19, 21, 23, 25, 27 or 29 °C. The maximum oocyst infection was observed at 25 °C with oocyst prevalence of 45% (95% CI: 23.1–68.5%) and mean oocyst intensity of 5 (95% CI: 1.6–8.4; Supplementary Fig. 1a). However, little or no infection was observed at 17 °C and 21 °C, and sporozoite infection was low at the temperature extremes (Supplementary Fig. 1b). In the next two experimental infectious feeds (referred to herein as Feed1 and Feed2),

gametocytemia in the blood meal and the mosquito sample size were increased to obtain more robust infection data. In Feed1, gametocytemia was adjusted to approximately 0.126% in the blood meal and the fed mosquitoes were kept at 17, 19, 21, 24, 25, 27, or 29 °C throughout. In Feed2, gametocytemia was increased to 0.139% and the blood-fed mosquitoes were kept at 18, 19, 20, 23, 25, 28, 29, and 30 °C. In these two feeds, maximum oocyst prevalence was observed at 23 and 25 °C for Feed1 (83.3%) and Feed2 (83.7%), respectively. For both feeds the mean oocyst intensity peaked at 23 °C, with 23.5 (Feed1) and 54.0 (Feed2) oocysts per mosquito with any observed oocysts. The lowest infection was observed at 17 and 18 °C, with oocyst prevalence of 13.3 to 25%, and mean oocyst intensities of 1.3 and 2.4 oocysts per oocyst-positive mosquito for Feed1 and Feed2, respectively (Supplementary Fig. 1a). Sporozoite prevalence estimates from Feed1 & Feed2 are presented in Fig. 1. There was little effect of parasite load on EIP in a preliminary study[34], which is also consistent with another recent report[35], so data from Feed1 & Feed2 were combined when modelling the EIP.

### Extrinsic Incubation Period

Sporogony varies between individual *A. gambiae* mosquitoes exposed to the same temperature, which can be represented by a probability distribution with a mean and variance. It's not possible to estimate the EIP of individual mosquitoes if they are dissected, but the cumulative increase in sporozoite prevalence with days post infection (DPI) is indicative of the cumulative distribution function (CDF), which represents the EIP variation across specimens. We estimate temperature-dependent changes in the EIP distribution and the human-to-mosquito-transmission probability (HMTP; the conditional probability of infection given the mosquito is alive) by fitting a mechanistic model of sporogony to the individual mosquito dissection data (oocyst intensity [counts] and sporozoite presence [1: any salivary gland sporozoites; 0: none]) from Feed1 and Feed2, in a Bayesian framework using Stan's Markov Chain Monte Carlo (MCMC) sampler. We fitted two types of model: an *independent* model, based on models fitted separately to data from individual temperatures; and a *pooled* model, based on a single model fitted to data from all temperatures, where certain model parameters vary as a function of temperature. In Fig. 1a, b, we show these model fits to sporozoite prevalence and oocyst intensity data (among mosquitoes with observable oocysts), respectively. Except for sporozoite prevalence at 17 °C, the actual vs fitted plots and $R^2$ or Brier skill scores indicate the model reasonably predicts the sampled values (Supplementary Fig. 2).

Results indicate that the EIP decreases as temperature increases (Fig. 2a). Our new EIP model enables us to describe the temporal variation in sporogony among mosquitoes at a given temperature: the time for 10%, 50% (the median) and 90% of the mosquito population to become infectious are denoted by $EIP_{10}$, $EIP_{50}$ and $EIP_{90}$. Consistent with the degree-day model, we see a non-linear increase in the EIP at lower temperatures. $EIP_{10}$, for example, increased from 7.6 days (95% Credible Intervals, CrI: 7.2–8.0) at 30 °C to 49.1 days (95% CrI: 44.2–55.1) at 17 °C (pooled model estimates) (Fig. 2a). The range between $EIP_{10}$ and $EIP_{90}$ decreased with increases in temperature: from 21.7 (95% CrI: 17.6–27.6) days at 17 °C to 3.7 (95% CrI: 3.4–4.0) days at 30 °C (Fig. 2a and Supplementary Fig. 3a). Similarly, the EIP distribution variance decreased with increases in temperature (Supplementary Fig. 3b). When the EIP standard deviation was normalised by the mean EIP there were little changes in this quantity with temperature, indicating that the variance scales with the mean (Supplementary Fig. 3c). The standard degree-day model, however, assumes all mosquitoes have the same EIP at a given temperature (Fig. 2a).

Using the pooled model fits, we also calculated the probability that our $EIP_{50}$ estimates are less than the degree-day model estimate: $p(EIP_{50} < EIP_D) = 0.95$ at approximately 24.6 days and $p(EIP_{50} < EIP_D) = 0.05$ at approximately 27.7 days (Fig. 2b). These

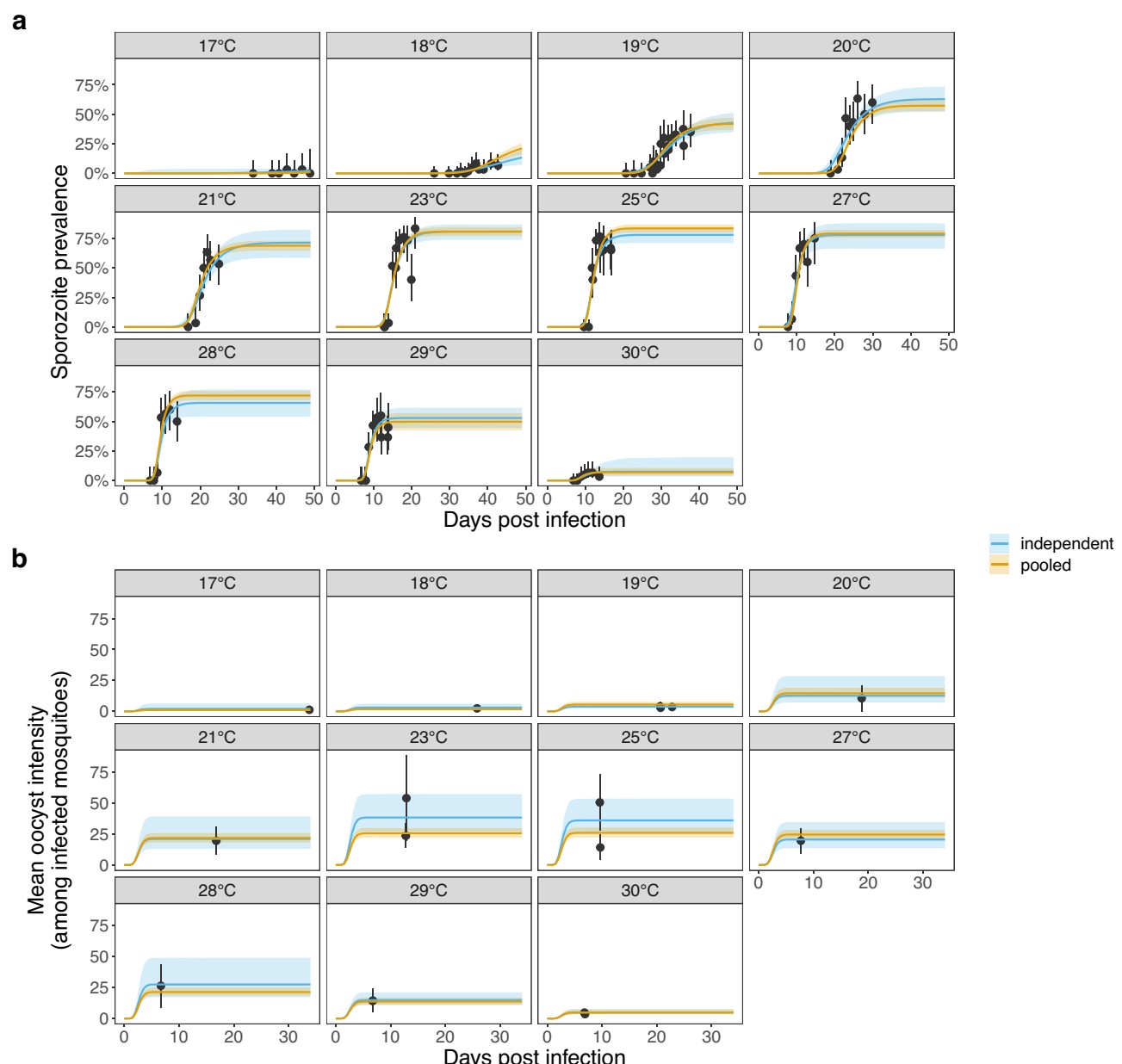

**Fig. 1 | Mechanistic model of sporogony fits sporozoite prevalence and mean oocyst intensity.** The temporal dynamics of sporogony is first fit to data from mosquitoes exposed to each constant temperature independently (independent model; blue lines). The model is also fit to all the standard membrane feeding assay mosquito dissection data simultaneously by fitting functional forms between temperature and certain model parameters (the independent model parameter estimates were used to guide the choice of functional forms) (pooled model; yellow lines). The actual (points) and predicted (lines) sporozoite prevalence and mean oocyst intensity are shown in (**a**) and (**b**) respectively. Uncertainty is shown by the 95% credible intervals of the posterior predictive means for the model estimates and 95% confidence intervals (error bars) for the experimental data, which was conducted with $n = 4195$ mosquitoes dissected for sporozoites (**a**) and $n = 520$ mosquitoes dissected for oocysts (**b**). For the binomial proportion data, confidence intervals were calculated using the Clopper-Pearson method. The facets indicate the constant temperature at which mosquitoes were maintained.

metrics indicate the established degree-day model provides a reasonable approximation of our EIP estimates at intermediate temperatures but becomes increasingly inaccurate at the temperature extremes. This finding is consistent with the original empirical data to which the degree-day model was fitted, which is also lower than the EIP predicted by the degree-day model at the lowest experimental temperature (Supplementary Fig. 4).

### Vector competence
The predicted mean oocyst load among infected mosquitoes ($\mu$) varied unimodally with temperature: the pooled model peaked at 24.5 °C with

a posterior value of 20.3 (95% CrI: 16.9–23.9) oocysts per mosquito (Fig. 3a). The maximum HMTP determined by the presence of any oocysts ($\delta_O$) varied unimodally with temperature: the pooled model value peaked at 24.8 °C with a posterior value of 83% (95% CrI: 80–86) (Fig. 3b). The probability that mosquitoes with any oocysts will mature any sporozoites ($\delta_S$; conversion of oocyst to sporozoite infection at the mosquito scale) is close to 100% at most of the temperatures investigated (Fig. 3c). At temperature extremes (and especially high temperatures), however, this value decreased: at 30 °C, for example, the pooled model posterior estimate is 14% (95% CrI: 9–20). As a result, the vector competence (we define as the maximum HMTP measured by

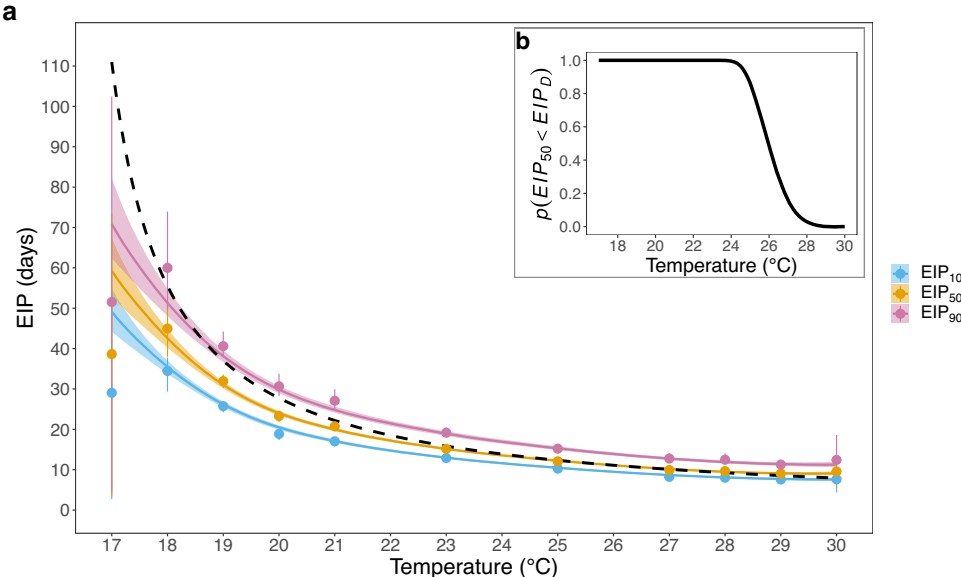

**Fig. 2 | The relationship between temperature and the EIP. a** The solid blue, yellow and pink lines show the relationship between temperature and pooled model $EIP_{10}$, $EIP_{50}$ and $EIP_{90}$ estimates respectively. Points show the equivalent independent model estimates. Data are presented as the median posterior values with the 95% credible intervals. The dashed black line shows the degree day model estimates. **b** shows the probability the pooled model posterior $EIP_{50}$ estimates are less than the corresponding degree-day model values ($EIP_D$), which were estimated using the 10,000 posterior MCMC values.

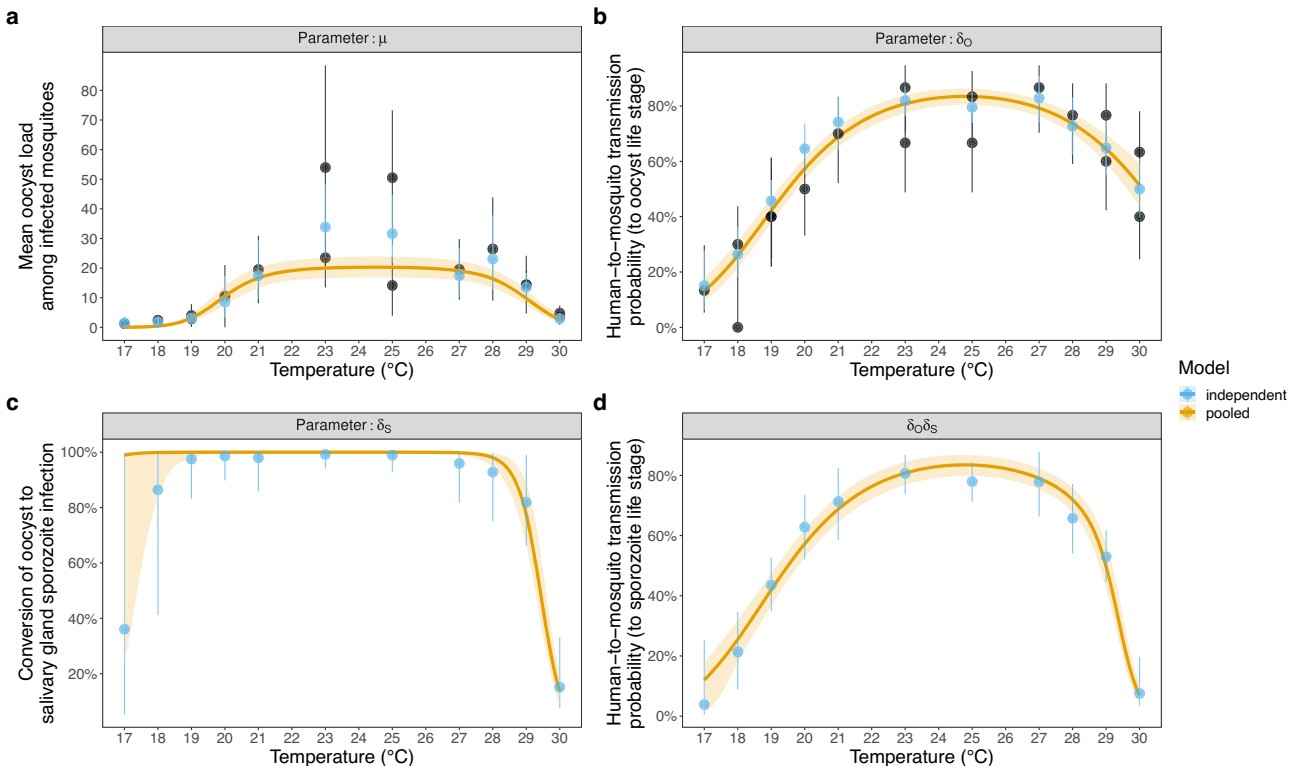

**Fig. 3 | The relationship between temperature and the fitted model mosquito infection parameter values. a** $\mu$ (the mean oocyst load among infected mosquitoes), **b** $\delta_O$ (the human-to-mosquito transmission probability, as determined by the presence of oocysts), **c** $\delta_S$ (the probability a mosquito infected with any oocysts will develop any sporozoites at the mosquito scale: conversion), and **d** $\delta_O\delta_S$ (vector competence; the human-to-mosquito transmission probability, as determined by the presence of sporozoites). For (**a**) and (**b**) black points show the sampled mean oocyst load and oocyst prevalence disaggregated by day of dissection with the 95% confidence intervals respectively, which were estimated from $n = 520$ dissected mosquitoes. All mosquitoes were dissected at sufficient days post infection that all oocysts were expected to have developed. For all plots blue points show the independent model estimates and yellow lines show the pooled model median posterior estimates with the 95% credible intervals.

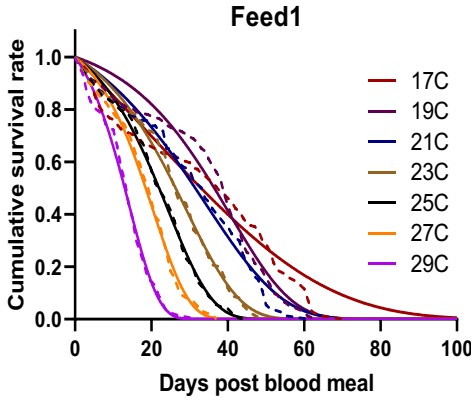

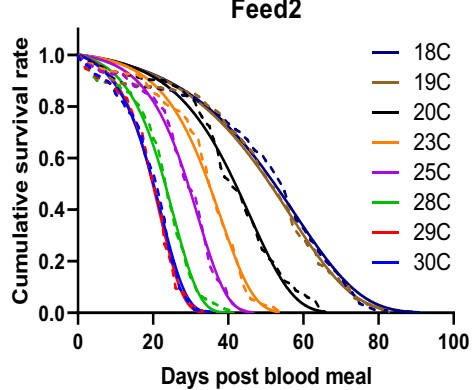

**Fig. 4 | Survival of *A. gambiae* mosquitoes fed with parasite-infected blood meals.** The cumulative survival was fitted with four representative survival models which were Gompertz, Weibull, loglogistic and negative exponential to determine best fit functions (see Supplementary Table 2 for model comparison).

Gompertz model (solid lines) best describes cumulative survival (dashed lines) of the mosquitoes (see Supplementary Table 3 for parameter values of Gompertz model). Source data are provided as a Source Data file.

the presence of any salivary gland sporozoites, $\delta_O \delta_S$) is unimodal and left-skewed, with a maximum at ~24.8 °C with a posterior value of 83% (95% CrI: 80–87; Fig. 3d).

### *A. gambiae* survival

Survival of mosquitoes fed with an infected blood meal was monitored across a range of temperatures from 17 to 30 °C. Cumulative survival was temperature-dependent (Fig. 4; Kaplan–Meier Log-Rank test for Feed1: $\chi^2 = 343.1$, df = 6, $P < 0.0001$; Kaplan–Meier Log-Rank test for Feed2: $\chi^2 = 1464$, df = 7, $P < 0.0001$), with median survival increasing as temperature decreased, at least across the temperature range studied (Supplementary Table 1). The cumulative survival data were best fitted by a Gompertz survival function, as indicated by the lowest corrected Akaike Information Criterion (AICc) values among four different survival models (Fig. 4, Supplementary Table 2 and Supplementary Table 3), which indicates the laboratory-based mosquitoes die at a faster rate as they age (i.e. survival is age-dependent). Survival was higher in Feed2 at all four temperatures (Kaplan–Meier Log-Rank; $\chi^2 = 109.3$, $P < 0.0001$ for 19 °C; $\chi^2 = 35$, $P < 0.0001$ for 23 °C; $\chi^2 = 32$, $P < 0.0001$ for 25 °C; $\chi^2 = 65.2$, $P < 0.0001$ for 29 °C).

### Implications for transmission and possible effects of climate warming

To explore the theoretical implications of our laboratory-derived parameter estimates for malaria transmission, we investigated how the Vectorial Capacity (VC) varied given recent and future mean temperatures for four locations in Kenya spanning a wide temperature range (Kericho, Kitale, Kisumu and Garissa, listed from coldest to warmest). Estimates of recent mean temperatures (1981–2000) and downscaled projections for future mean temperatures (2046–2065) for these locations were obtained from previous malaria-climate research[36] (Table 1). VC is a common transmission metric that describes the number of infectious mosquito bites arising per infectious human per day assuming complete transmission efficiency and a susceptible human population[37]. VC is related to the basic reproduction number, $R_0$, but comprises a subset of the parameters to provide a measure of the entomological transmission potential of a mosquito population. Absolute VC estimates require measures of the vector-host ratio and how the vector-host ratio could change with climate, which are unknown for our locations. We therefore restrict our analysis to vector traits we have estimates for and calculate the expected number of infectious bites per a single mosquito, which we denote as the relative Vectorial Capacity: rVC. In the original derivation of VC, the per-capita mosquito mortality rate is assumed to be constant meaning

the age when a mosquito is infected does not affect the results. Given the observed age-dependent mosquito mortality, the age at which a mosquito is infected can affect the number of infectious bites made per mosquito[38]. We therefore apply our temperature-dependent life history traits (EIP, age-dependent mosquito survival and biting rate) to estimate the expected number of infectious bites per infected mosquito, $z$, given it is infected at age $t$ by adapting the methods developed by Iacovidou et al.[38] and conduct a sensitivity analysis for the age at which mosquitoes are infected (Supplementary Fig. 5). The sensitivity analysis indicates that adult mosquitoes infected at $t_0$ (i.e., at the youngest age upon emergence) generate the maximum number of infectious bites (Supplementary Fig. 5a), so we give estimates of $\mathbb{E}(z|\text{infected at age } t_0)$. This quantity does not account for vector density, but if density remains unchanged then the proportional difference in rVC estimates between EIP models and/or over time is equivalent to the proportional difference in absolute vectorial capacity within a given location. To propagate uncertainty in the EIP and biting rate estimates, we calculate rVC by sampling posterior parameters and estimating the EIP using our pooled model or the existing degree-day model using the recent and future mean temperatures for the four locations in Kenya. This approach enables us to compare the transmission consequences of our novel measures of EIP with the equivalent estimates from the degree-day model. Given that we did not measure biting rate in our current study, we estimated temperature-dependent changes in the posterior biting rates for another malaria vector, *A. stephensi*, using data presented in[24] (Supplementary Fig. 6) and sampled from these posteriors to calculate the rVC. Further methods and summary of the analysis are provided in Supplementary Fig. 7 and Supplementary Table 4 for determining survival distributions.

The predicted estimates for rVC using values of EIP derived from our model and the degree-day model for each of the locations in Kenya are presented in Table 1 (upper). Given there were differences in adult mosquito survival between Feed1 and Feed2, we present results for these experiments separately. Under recent climate conditions in Kericho, which is the coldest environment of the four locations, the rVC using our EIP model was 13 ($\frac{0.38}{0.03}$; see these rVC values in Table 1 upper) and 26 ($\frac{0.79}{0.03}$) times higher than that of the degree-day model for Feed1 and Feed2, respectively (Table 1 upper). Under warmer conditions the EIP models converge, so the differences in predictions of rVC become progressively smaller from Kitale to Kisumu, regardless of feed. As temperatures increase further, our EIP estimates are shorter than the degree-day model, so for the warmest location, Garissa, the predicted rVC was 0.92 for Feed1 and 0.95 for Feed2 of the equivalent values derived using the degree-day model. For the future climates, the

**Table 1 | Influence of Extrinsic Incubation Period (EIP) on recent and future vectorial capacity for four locations in Kenya**

Relative vectorial capacity (rVC)

| Year | Location (mean temperature °C) | Biting rate (95% CrI) | $\mathbb{E}(z\|\text{infected at age}_0)$ (95% CrI) | | | |
| --- | --- | --- | --- | --- | --- | --- |
| | | | Feed1 | | Feed2 | |
| | | | Suh-Stopard | Degree-day | Suh-Stopard | Degree-day |
| Recent (1981–2000) | Kericho (17.5) | 0.126 (0.09–0.156) | 0.38 (0.26–0.51) | 0.03 (0.02–0.03) | 0.79 (0.53–1.05) | 0.03 (0.02–0.03) |
| | Kitale (19) | 0.154 (0.124–0.182) | 1.19 (0.95–1.37) | 0.77 (0.62–0.9) | 2.34 (1.87–2.7) | 1.67 (1.35–1.95) |
| | Kisumu (23.4) | 0.243 (0.219–0.269) | 2.57 (2.32–2.85) | 2.53 (2.32–2.79) | 4.57 (4.15–5.08) | 4.55 (4.17–5.01) |
| | Garissa (28.8) | 0.345 (0.315–0.378) | 1.79 (1.65–1.99) | 1.94 (1.79–2.14) | 3.79 (3.49–4.19) | 3.99 (3.67–4.41) |
| Future (2046–2065) | Kericho (19.5) | 0.164 (0.135–0.192) | 1.49 (1.22–1.7) | 1.1 (0.92–1.27) | 2.85 (2.32–3.24) | 2.26 (1.88–2.62) |
| | Kitale (20.9) | 0.192 (0.166–0.219) | 2.15 (1.84–2.4) | 1.89 (1.62–2.12) | 3.87 (3.31–4.33) | 3.55 (3.03–3.97) |
| | Kisumu (25.2) | 0.279 (0.254–0.307) | 2.45 (2.25–2.7) | 2.49 (2.3–2.73) | 4.53 (4.16–4.96) | 4.58 (4.23–5.03) |
| | Garissa (30) | 0.363 (0.331–0.398) | 1.42 (1.28–1.63) | 1.74 (1.6–1.9) | 3.3 (3–3.72) | 3.74 (3.45–4.11) |

**Fold change in the rVC (of mosquitoes infected at age 0) due to predicted future mean temperatures (ratio of future:recent) using estimates of EIP from either the current study or the established degree-day model (95% CrI)**

| Location | Feed1 | | Feed2 | |
| --- | --- | --- | --- | --- |
| | Suh-Stopard | Degree-day | Suh-Stopard | Degree-day |
| Kericho | 3.93 (3.12–5.34) | 43.09 (40.82–49.65) | 3.63 (2.88–4.96) | 79.02 (75.2–88.79) |
| Kitale | 1.81 (1.69–2) | 2.44 (2.34–2.72) | 1.66 (1.57–1.83) | 2.11 (2.02–2.36) |
| Kisumu | 0.95 (0.93–0.99) | 0.98 (0.96–1.02) | 0.99 (0.96–1.03) | 1 (0.98–1.05) |
| Garissa | 0.8 (0.76–0.84) | 0.89 (0.88–0.91) | 0.88 (0.85–0.91) | 0.93 (0.92–0.95) |

(Upper table) In the absence of vector-host ratio estimates for different field settings, relative vectorial capacity (rVC) is represented by the expected number of infectious bites per mosquito that is infected on emergence to adulthood (age zero) ($\mathbb{E}(z\|\text{infected at age}_0)$). We estimate this quantity using data from two experiments (Feed1 and Feed2), comparing the posterior estimates from our new model for EIP (Suh-Stopard) with the longstanding degree-day model. (Bottom table) Relative fold change in rVC, assuming no changes in the vector-host ratio, under climate warming is estimated by the ratio of the future:recent $\mathbb{E}(z\|\text{infected at age}_0)$ estimates for each EIP model. For all calculations, the model was run with 100 posterior samples for each parameter value, and the median and 95% quantiles (95% credible interval, CrI) were calculated from these values.

rVC predictions were qualitatively similar, but because temperatures are shifted right along the EIP curve, the differences between models in Kericho were smaller (rVC based on our EIP model was now just 1.35 and 1.26 times higher than the degree-day model for Feed1 and Feed2, respectively) and slightly larger in Garissa (rVC values using our EIP model were 0.82 that of the degree-day model for Feed1 and 0.88 for Feed2) (Table 1 upper, Supplementary Fig. 5a).

To explore the overall implications for climate change, we examine the relative difference in rVC using the ratio of rVC in recent and future conditions (Table 1 bottom). Differences between the ratio of future:recent rVC estimated using our EIP distributions and the degree-day model were consistently predicted irrespective of age of mosquito infection (Supplementary Fig. 5b). With the degree-day model, we estimate transmission intensity to increase 43.1 (40.8–49.6 95% CrI) (Feed1) and 79.0 (75.2–88.8 95% CrI) (Feed2) times in Kericho due to climate warming. Using our model of EIP, the increase in transmission is only 3.93 (3.12–5.34 95% CrI) (Feed1) and 3.63 (2.88–5.96 95% CrI) (Feed2) times. These results indicate that the degree-day model over-estimates increase in future transmission potential by 11 (Feed1: $\frac{43.1}{3.93}$) to 22 times (Feed2: $\frac{79.0}{3.63}$) relative to our EIP model. For the other locations, there is much better agreement between EIP models (Table 1 bottom).

## Discussion

For over 60 years, the development time of the most deadly human malaria-causing parasite, *P. falciparum*, in anopheline mosquitoes has been approximated using a single fit of a degree-day model[13]. In this study, we generated sporogony data for *P. falciparum* in a dominant African vector species, *A. gambiae*. These data were then analysed using a mechanistic model of sporogony to fully characterise the kinetics of parasite development across a range of constant temperatures. Our $EIP_{50}$ estimates align well with the established degree-day model EIP estimates for constant temperatures between approximately 24 to 28 °C (Fig. 2). Outside this temperature range, the degree-day model estimates are not within the 95% CrIs of $EIP_{50}$; at 17 °C, for example, while the uncertainty of the pooled model was relatively large because the low infection rates resulted in poor model fitting, we estimated $EIP_{50}$ as 59 days with a 95% CrI of 53-68 days, whereas the degree-day model predicts 111 days. The reasons for the marked differences between our EIP estimates and those of the degree-day model at cooler temperatures are unclear. The initial study of Nikolaev[23] that was used to fit the degree-day model did not measure EIP below 19-20 °C so it could be a true difference between the experimental systems, a consequence of the degree-day model functional form, or because the model was fitted to limited data. Studies in other systems suggest non-linear effects of temperature under cool conditions which could violate the linear growth rate assumption of the degree-day model[39,40], and also contribute to the relatively poor fit of our mechanistic model at the thermal limits. There could also be differences between mosquito species, or even populations. Preliminary studies suggest there can be differences in thermal performance of vector competence and EIP between species of malaria vector infected with the same parasite strain[25,41]. Our data suggest sporogony can be completed more rapidly in *A. gambiae* at cooler temperatures than predicted by the Detinova degree-day model but other data (albeit limited) on *An. elutus*, suggest longer EIP than the Detinova model at warmer temperatures[42]. Yet the prevailing assumption within mathematical models has been that the EIP is a property of the parasite species and temperature alone[7]. Future studies on other key malaria vectors are therefore important to further determine the role of vector species.

Consistent with other recent studies[24,25], our research highlights variation in the EIP among mosquitoes exposed to the same environment. The mechanisms that generate this variation are unclear but could include intraspecific variation of individual mosquito characteristics such as body size[43], nutritional condition[44], or density-dependent effects among individual parasites[45–47]. This biological variation challenges a classic assumption of mechanistic malaria transmission models that typically assume all mosquitoes become infectious at the same time point following an infectious blood meal[37], and highlights the methodological requirement to sample sufficient mosquitoes over time to enable estimation of the mean or median EIP ($EIP_{50}$).

In addition, our study provides insights into the temperature-dependence of the HMTP as measured by the presence of oocysts and sporozoites (vector competence). The maximum prevalence of oocysts and the mean number of oocysts per mosquito both had a unimodal relationship with temperature, with much less parasite establishment at thermal extremes. These findings support other experimental infection studies suggesting the earlier stages of parasite development are most temperature sensitive[27,28,48]. Our vector competence estimates also demonstrate a unimodal relationship, with an optimum of 24.8 °C. The asymmetric nature of this relationship appears to result in part from a strong effect of higher temperatures on the probability that an infected mosquito will develop sporozoites. To our knowledge, this is the first parameterisation of conversion efficiency of oocyst to sporozoite-infection across different constant temperatures. The conversion efficiency is 100% at most temperatures, meaning oocyst prevalence is a good measure of the prevalence of infectious mosquitoes. Many studies assume that infection prevalence at the oocyst stage (percent infected) can be used as a proxy for infection prevalence at the sporozoite stage (percent infectious)[27,49–51]. Our data suggest this relationship is not robust at higher or possibly lower temperatures and could be an important factor determining the edges of the thermal range for transmission. At high temperatures, for example, our laboratory strains showed a precipitous decline of this conversion efficiency to 14% between 28 and 30 °C (Fig. 3c), meaning at these temperatures the presence of oocysts might not be an accurate measure of infectivity. Accurately quantifying sporozoite load is challenging, so we measured sporozoite prevalence, meaning we could only model this conversion efficiency at the mosquito scale. Understanding conversion probability and the mechanisms involved at the parasite scale will require further experiments and model fitting, ideally with field-derived sympatric parasite and mosquito strains.

Finally, there is limited research on the effects of temperature on *A. gambiae* survival, particularly towards the lower thermal limits of transmission, or including the influence of infectious blood meals[52]. Our data show that across the temperature range studied, survival increases as temperatures decrease, though at even cooler temperatures it is likely that survival will decrease[9,53]. Given the controlled insectary conditions, the differences in survival between Feed 1 & 2 were surprising and could suggest possible interactions between temperature and blood meal quality (different batches of blood, human serum etc.) or parasite load within the blood. Regardless, survival was best characterised using Gompertz functions, consistent with age-dependent mortality. An increasing number of studies highlight the importance of considering age-dependent mortality[38,54–58], although most transmission models assume that mortality remains constant with age. The extent of age-dependent mortality in the field where mosquitoes are subject to greater causes of daily mortality (e.g. predation, swatting during blood feeding, stress associated with searching for oviposition sights, control tools) remains an important question[58,59].

Incorporating our data into a transmission metric that captures variability in the EIP and age-dependence of mosquito mortality reveals important implications for understanding current and future transmission intensity. Across the four locations in Kenya, rVC is low in Kericho, which approaches the observed lower thermal limit for

transmission, increases through Kitale and Kisumu which are at intermediate temperatures, and then declines slightly in Garissa, which is towards the upper limits for transmission. This pattern is consistent with a previously identified unimodal thermal performance curve for transmission[9,60]. However, our shorter EIPs result in higher rVC than predicted using estimates of EIP from the established degree-day model under cooler conditions. This result suggests that all else being equal, models of transmission that use the standard predictions of EIP are likely to underestimate transmission intensity in low temperature environments where differences between EIP models are greatest. At the other extreme, because there is a cross-over in the EIP models, our results suggest that conventional models will tend to marginally overestimate transmission at high temperatures. The unimodal relationship between temperature and transmission is present using either model of EIP, which means that the effects of climate change are predicted to be 'two-tailed'[41,61,62], with transmission potential increasing in cooler environments, remaining largely unchanged at intermediate temperatures, and declining in warm environments. However, the predicted relative increase in transmission in Kericho was 11-22 times greater when using the degree-day estimates of EIP than predicted by our new estimates of EIP. Numerous studies predict variation in the dynamics and distribution of malaria due to climate warming, with increases in transmission of particular concern for colder environments such as the Kenyan Highlands[22,30,63–67]. Our results suggest that current predictions of the impact of climate change likely overestimate the increase in transmission risk in cooler areas because the Detinova degree-day model predicts both slower rates of parasite development under current conditions, and a greater rate of change in response to future warming, relative to our model. The quantitative effect could vary depending on other factors such as the exact pattern of mosquito mortality in the field or changes in vector density, but because our temperature-dependent EIP model is flatter than the degree-day model at the cooler end of the scale, we expect the overestimation to be qualitatively robust. Given the potential public health significance, these insights highlight the need to understand the effects of temperature at the fringes of transmission.

As in the original empirical work used to parameterise the Detinova degree-day model, our study manipulated temperatures experienced by the adult mosquitoes as these are known to affect parasite development rate directly. Yet, larval rearing conditions can have indirect transstadial effects that can further influence vector competence[68–70] and parasite development[44]. Thus, it could be valuable to extend research to include variation in larval conditions, such as temperature and resource availability, as well as variation in adult conditions. It should be noted, however, that temperatures in larval habitats need not be the same as those experienced by the adults[71] and adult mosquitoes in a single feeding and resting location could in principle derive from diverse larval habitats, adding complexity to both experimental design and interpretation.

A further potential limitation of our study is that we considered only one strain of *A. gambiae* and one strain of *P. falciparum*, both of which have been maintained under controlled lab settings for many generations. However, there could be intrinsic differences between strains[35], or differences as a result of adaptation to local environments; at present, there is limited research on the extent of thermal adaption in mosquitoes or parasites, but in principle this could shape the influence of temperature on local sympatric mosquito and parasite pairings[72]. Other factors could also be important, such as the influence of multiple blood meals on EIP[73,74], the effects of daily temperature fluctuation in addition to mean temperature[25,28,71], and possible effects of parasite load on infection kinetics in the mosquito[33] (but see[35], which showed no effect). Nonetheless, our highly resolved sporogony data, together with data on the temperature-dependence of vector competence and mosquito survival, provide new insights that challenge current understanding. Our findings highlight the importance of better

characterising the thermal dependence of mosquito and parasite traits.

## Methods
### Mosquitoes
*Anopheles gambiae* (G3 strain) mosquitoes were used throughout the experiments. Mosquitoes for all developmental stages were reared under standard insectary conditions at 27 °C ± 0.5 °C, 80% ± 5% relative humidity, and a 12 h:12 h light-dark photoperiod. Larval density and amount of larval food (ground TetraFin™; Tetra, Blacksburg, VA) were standardised to ensure uniform adult size. Adult mosquitoes were maintained on 10% glucose solution supplemented with 0.05% para-aminobenzoic acid (PABA). For the infectious feeds, 5-6-day-old female mosquitoes were randomly aspirated into cardboard cup containers covered with netting and starved for approximately 6 h before the infectious feed. Individual containers contained 150-200 mosquitoes. All experiments were conducted under Penn State Institutional Biosafety Committee protocol # 48219, which covers protocols for handling mosquitoes during rearing, blood feeding and infection. We used blood from a commercial biospecimen supplier (BioIVT, corp) that derived from de-identified human donors and was not collected specifically for our study.

### Mosquito transmission and survival studies
In vitro cultured *Plasmodium falciparum* (NF54 isolate, MR4) was provided by the Parasitology Core Lab (http://www.parasitecore.org/) at Johns Hopkins University as described previously[28]. In brief, gametocyte culture in stage four to five (day 14 after gametocyte initiation) was transported to Penn State overnight, and gametocyte-infected erythrocytes were maintained > 24 h before the infectious feed to allow additional maturation of gametocytes.

Mosquitoes were fed on day 16 post gametocyte initiation. The proportion of erythrocytes infected with mature gametocytes (i.e., gametocytemia) generally ranged between 1–3% in the culture. An infectious blood meal was prepared by mixing gametocyte-infected erythrocytes with fresh human serum and erythrocytes at 40% haematocrit on the day of blood feeding as previously described[28].

All infectious feeds were conducted in a walk-in environment-controlled chamber. Glass bell jars were uniformly covered with Parafilm to serve as membrane feeders and heated to 37 °C with continuously circulating water as described in[44]. Containers of mosquitoes were randomly allocated to bell jars to minimise any effect of position or feeder. Mosquitoes were fed for 20 min at 27 °C after acclimating at 27 °C for an hour, and > 95% mosquitoes were fully engorged in all infectious feeds. Immediately after blood feeding, mosquitoes were placed into incubators (Percival Scientific Inc., Perry, Iowa) with appropriate temperature treatment conditions (90% ± 5% relative humidity, and 12 h:12 h light-dark photoperiod) and provided daily with fresh 10% glucose solution supplemented with 0.05% PABA.

Approximately 300 to 1200 mosquitoes in two-to-six containers (150 to 200 in each container) were fed infectious blood meals depending on temperature treatment in three independent experiments. In each experiment, seven to eight temperatures were selected between 17 °C and 30 °C as these temperatures span the temperature range for *P. falciparum* infection to be observed in previous studies[25,28]. Two infectious feed experiments (Feed1 & Feed2) were conducted after an initial pilot trial. In the pilot study, gametocytemia of parasite-infected blood meal was 0.024% and blood-fed mosquitoes were kept at constant temperatures of 17, 19, 21, 23, 25, 27, and 29 °C throughout. This pilot generated little or no infections at lower temperatures. In the main two infectious feeds (i.e., Feed1 and Feed2) gametocytemia and mosquito sample size were increased. In Feed1, gametocytemia of blood meals was adjusted to 0.126% and blood-fed mosquitoes were kept at constant temperatures of 17, 19, 21, 23, 25, 27, and 29 °C throughout. In Feed2, gametocytemia was adjusted to 0.139% and

blood-fed mosquitoes were kept at constant temperatures of either 18, 19, 20, 23, 25, 28, 29, or 30 °C to further increase the sample size of infected mosquitoes for low and high temperatures. The air temperature of the incubators (Percival Scientific Inc., Perry, Iowa) was monitored closely using HOBO data loggers (Onset Computer Corporation, Bourne, MA; error range = ± 0.1 °C) at 15 min intervals, and the accuracy of temperature was maintained at ±0.2 °C. Data loggers and incubators were rotated throughout all infectious feed experiments to minimise potential equipment effects. The number of mosquitoes and containers varied by experiments as presented in Supplementary Table 5. In general, sample size of mosquito dissections and number of containers were increased for low and high temperatures to ensure sufficient sporozoite infections to estimate the EIP distribution.

To determine parasite infections, mosquitoes were randomly collected by aspirating into 95% ethanol, and midguts and salivary glands were dissected in 1× phosphate-buffered saline solution under a standard dissecting scope as previously[25]. Presence or absence of parasite infection was determined by examining midguts for oocysts or salivary glands for sporozoites using a compound microscope. Visible oocysts in midguts were counted. To ensure correct scoring, oocysts and sporozoites were inspected under 40× magnification and cross-checked by a second person. Oocyst or sporozoite prevalence was calculated as the total number of infected mosquitoes divided by the total number of dissected mosquitoes by combining dissection data from replicated containers of mosquitoes for each temperature treatment. Oocyst intensity and prevalence was determined on the days when maximum infection is expected based on previous studies[25]. To determine the EIP, ~10–60 salivary glands of mosquitoes were dissected daily (10–15 per container). Dissection days were determined to capture the first sporozoite infection followed by increase in prevalence to maximum within each temperature treatment. Detailed sample size and dissection days are reported in Supplementary Table 5.

To obtain survival data, dead mosquitoes were scored daily in Feed1 and Feed2 experiment until all mosquitoes were dead. The mosquito containers were rotated daily within each incubator throughout the experiment to minimise potential effect of container position on survival.

### Estimating the Extrinsic Incubation Period and human-to-mosquito transmission probability

**Mechanistic model of sporogony.** Visualising the developmental stage of *Plasmodium* parasites typically requires mosquito dissection (but see ref. 75 for alternative experimental methods to determine sporozoite infection). The EIP of individual mosquitoes cannot be estimated from mosquito dissection data because dissection kills the mosquito. Consequently, the EIP is typically estimated among the mosquito and parasite populations as a whole using a statistical or mechanistic model[33]. In previous work, we demonstrated how the cumulative increase in sporozoite prevalence in standard membrane feeding assays with time, $n$, can, however, be used to estimate the CDF (F($n$)) of EIPs among individual mosquitoes using a mechanistic model that simulates the development times of individual parasites within the mosquito[33]. To estimate the EIP, we modified and fitted this model[33]. The model consists of five key parameters: the HMTP (as determined by the presence of oocysts; $\delta_O$), the parasite load (as determined by the number of oocysts per mosquito) distribution, which is described by the mean ($\mu$) and overdispersion ($k$) parameters of a zero-truncated negative binomial distribution, and the parasite development time parameters (shape, $\alpha$, and rate, $\beta$, of the gamma distribution). These development time parameters are constituted by the sum of two independent development times from inoculation ($G$) to first oocyst appearance ($O$) and from $O$ to salivary gland

sporozoites ($S$). This model allows the estimation of two quantities related to parasite development times: (1) the distribution of individual parasite development times from inoculation to sporozoite (parasite numbers are equivalent to the number of oocysts) and (2) the distribution of development times for individual mosquitoes to have completed sporogony (EIP distribution). According to this model the CDF of the EIP distribution, $n$, is calculated as,

$$F(n) = \frac{1 - k^k (k + \mu Q(\alpha,0,\beta n))^{-k}}{1 - \left(\frac{k}{k+\mu}\right)^k}, \tag{2}$$

where $Q(\alpha,0,\beta n) = \frac{\Gamma(\alpha,0,\beta n)}{\Gamma(\alpha)}$ (the regularised gamma function), $\Gamma(\alpha) = \int_0^\infty t^{\alpha-1} e^{-t} dt$ (the complete gamma function) and $\Gamma(\alpha,0,\beta n) = \int_0^{\beta n} t^{\alpha-1} e^{-t} dt$ (the generalised incomplete gamma function). The modelled sporozoite prevalence at time, $n$, therefore depends on the proportion of infected mosquitoes that are infectious (F($n$)) and the proportion of mosquitoes that are infected. *P. falciparum* infection of *A. gambiae* is unlikely to increase mosquito mortality because the parasite is highly adapted to this sympatric mosquito vector[41,76], and examination of our raw data provided no evidence of a decline in sporozoite prevalence observed over time. We therefore assumed the relative survival of infected to uninfected mosquito parameter, which fitted in[33], was equal to 1. To facilitate model fitting, an additional parameter that estimates the probability a mosquito infected with oocysts will develop any observable sporozoites ($\delta_S$) was incorporated at the mosquito scale to account for complete clearance of oocyst infection or complete lack of oocyst bursting, which was observed for certain temperature treatments. It would be preferable to account for this observation at the parasite scale: specifically, the loss of individual oocysts would be modelled, however, sporozoite load data was unavailable meaning it was not possible to parameterise this model.

We used a binomial likelihood to fit the probability of observing a mosquito with salivary gland sporozoites such that,

$$I_n \sim \text{Bin}\left(D_n, \delta_O \delta_S F(n)\right), \tag{3}$$

where $I_n$ is the number of infectious mosquitoes at days post infection $n$ and $D$ is the number of dissected mosquitoes at time $n$. We calculated the likelihood of observing a mosquito dissected with $Y$ oocysts at time $n$ as,

$$\Pr(Y_n | \Gamma_{GO}(n), \delta_O, n, \mu, k)$$

$$= \begin{bmatrix} (1-\delta_O) + \delta_O \left( \dfrac{\left(\frac{k}{k+\mu}\right)^k - \left(\frac{k}{k+\Gamma_{GO}(n)\mu}\right)^k}{-1 + \left(\frac{k}{k+\mu}\right)^k} \right), if\ Y = 0 \\[4mm] \delta_O \dfrac{\left( k^k (\Gamma_{GO}(n)\mu)^{Y_n} (k + \Gamma_{GO}(n)\mu)^{(-Y_n-k)} \binom{-1+Y_n+k}{-1+Y_n} \right)}{1 - \left(\frac{k}{k+\mu}\right)^k}, if\ Y \geq 1, \end{bmatrix} \tag{4}$$

where $\Gamma_{GO}$ is the CDF of the individual parasite development times from $G$ to $O$. The appearance of early oocysts was not available for these data as mosquitoes were all dissected when oocysts were late stage. This means that the distribution of transition times between $G$ and $O$ (and its dependence on temperature) was unidentifiable. We therefore specified informative priors on $\alpha_{GO}$ and $\beta_{GO}$ such that the modelled oocyst prevalence peaked prior to the days post infection mosquitoes were dissected for oocysts. The mean prior values for $\alpha_{GO}$ and $\beta_{GO}$ were determined by the corresponding values obtained from previously published model fits[1].

**Modelling the effects of temperature on sporogony.** To identify suitable functional forms for the relationship between model

parameters and temperature, first, we fit the model to data from mosquitoes exposed to each constant temperature independently. The independent model parameter estimates were then used to guide functional relationships between temperature and the model parameter values, which were estimated by fitting the model and temperature-dependent functions to all data (pooled model). All model fits were checked visually. To incorporate differences in development time from $G$ to $S$, we allowed $\alpha_{OS}$ and $\beta_{OS}$ to vary with temperature. In doing so, the model aims to estimate the complete development time from $G$ to $S$, but not the constituent times from $G$ to $O$ and $O$ to $S$. We fitted the model with the following functional relationships between the model parameters and temperature ($T$):

$$
\begin{aligned}
\alpha_{OS}(T) &= a_\alpha T^2 + b_\alpha T + c_\alpha, \\
\beta_{OS}(T) &= m_\beta T + c_\beta, \\
\delta_O(T) &= \frac{1}{1 + e^{-(a_O T^2 + b_O T + c_O)}}, \\
\delta_S(T) &= \frac{1}{1 + e^{-(a_S T^2 + b_S T + c_S)}}, \\
\mu(T) &= \frac{d}{1 + e^{-(a_\mu T^2 + b_\mu T + c_\mu)}}.
\end{aligned}
\tag{5}
$$

We fitted both the pooled and independent models to experimentally infected *A. gambiae* mosquitoes that were dissected for oocysts number or salivary gland sporozoites on different days post infection in a Bayesian framework using the No U-turn Markov-chain Monte Carlo (MCMC) sampler[77], with four chains each with 5500 iterations (inclusive of 3000 iterations warmup). Convergence is determined by $\hat{R} < 1.01$ for all parameters. Temperature was scaled so that the mean temperature was 0 and its standard deviation was 1. To account for uncertainty in the parameter estimates for all quantities, we calculate the estimates for each MCMC sample and give the median and 95% credible intervals (CrI) of these values. Independent and pooled model priors are given in Supplementary Table 6.

**Quantifying the EIP distribution.** Differentiating $F(n)$ we obtain the EIP probability distribution function (PDF; $f(n)$):

$$
f(n) = \frac{\beta e^{-\beta n}(\beta n)^{\alpha-1} k^{1+k} \mu (k + \mu Q(\alpha, 0, \beta n))^{-1-k}}{\left(1 - \left(\frac{k}{k+\mu}\right)^k\right) \Gamma(\alpha)}
\tag{6}
$$

The time for different percentiles, $q$, of the mosquito population to have completed the EIP (denoted by $EIP_q$) is given by the inverse CDF ($F^{-1}(t)$), which is calculated thus:

$$
F^{-1}(q) = \frac{QI\left(\alpha, 0, \left(\left(\frac{-q(1-(k/(k+\mu))^k)-1}{k^k}\right)^{-1/k} - k\right)/\mu\right)}{\beta},
\tag{7}
$$

where QI is the inverse of the regularised gamma function.

The mean and variance of the EIP distributions were estimated at each temperature by numerical integration using the Cuhre algorithm (R calculus package[78]): $\mathbb{E}[n] = \int_0^\infty n f(n)\, dn$ and $Var(n) = \mathbb{E}[n^2] - (\mathbb{E}[n])^2$ respectively.

## Determining survival distribution of *A. gambiae* mosquitoes fed with parasite-infected blood meals

To characterise the mosquito survival distribution, the cumulative survival was fitted with four survival models: Gompertz, Weibull, loglogistic and negative exponential. These models have been used to describe survival curves of a diversity of insects in laboratory and field settings[56,79–84]. The corrected Akaike Information Criterion (AICc) value for these models were compared, indicating the Gompertz model is the best fit for all temperature treatments in Feed1 and Feed2. The equation used for Gompertz model was,

$$
S_g(x) = e^{\frac{a}{b}(1 - e^{bx})}
\tag{8}
$$

where $S_g(x)$ is cumulative proportion surviving at age $x$, $a$ is the initial mortality rate, and $b$ is the age-dependent mortality rate[84–86].

## Estimating the effects of EIP on expected number of infectious bites per infected mosquito under recent and future temperature conditions

To estimate the effects of the different EIP estimates on the relative vectorial capacity, we adapted the methods developed by Iacovidou et al.[38] to calculate the expected number of infectious bites per infected mosquito, $z$, given the mosquito age at infection and a Gompertz survival model. In all cases, $n$ is the EIP; $t_0$, $t_1$ and $t_2$ are the mosquito ages at infection, at completion of the EIP and when death occurs, respectively; $a$ and $b$ are the Gompertz survival model parameters; $B$ is the per-mosquito biting rate per day; $j$ is the number of bites. In the original methods the probability a mosquito survives the EIP given it is infected at age $t_0$ is calculated thus,

$$
\mathbb{P}\left(\text{surviving EIP}|\text{infected at age } t_0\right) = \int_{n=0}^\infty [f(n)] e^{-\frac{a}{b} e^{t_0 b}(e^{nb}-1)}\, dn.
\tag{9}
$$

The probability a mosquito bites $z$ times given it completes the EIP at age $t_1$ is then calculated,

$$
\mathbb{P}\left(z = j | \text{completes EIP at age } t_1\right) = \int_{t_2 = t_1}^\infty \frac{B^j (t_2 - t_1)^j}{j!} e^{t_2 b - (t_2 - t_1)B} \\
a e^{-\frac{a}{b}(e^{t_2 b} - e^{t_1 b})}\, dt_2.
\tag{10}
$$

These equations are combined to estimate the probability a mosquito bites $z$ times given it is infected at age $t_0$,

$$
\mathbb{P}\left(z = j | \text{infected at age } t_0\right) = \begin{cases} \int_{t_1 = t_0}^\infty \mathbb{P}(z = j | \text{completes EIP at age } t_1) [f(t_1 - t_0)] e^{-\frac{a}{b} e^{t_0 b}(e^{(t_1 - t_0)b} - 1)} dt_1 & \text{if } j \neq 0, \\ \int_{t_1 = t_0}^\infty \mathbb{P}(z = j | \text{completes EIP at age } t_1) [f(t_1 - t_0)] e^{-\frac{a}{b} e^{t_1 b}(e^{(t_1 - t_0)b} - 1)} dt_1 + \left(1 - e^{-\frac{a}{b} e^{t_0 b}(e^{(t_1 - t_0)b} - 1)}\right) & \text{if } j = 0. \end{cases}
\tag{11}
$$

And the expectation of $\mathbb{P}(z = j | \text{infected at age } t_0)$ is calculated thus,

$$
\mathbb{E}(z | \text{infected at age } t_0) = \sum_{j=0}^\infty j\, \mathbb{P}(z = j | \text{infected at age } t_0)
\tag{12}
$$

We modified this model to calculate the equivalent quantity for the degree-day model, which assumes all mosquitoes have the same EIP:

$$
t_1 = t_0 + EIP_D,
\tag{13}
$$

$$
\mathbb{P}\left(\text{surviving EIP}|\text{infected at age } t_0\right) = e^{-\frac{a}{b} e^{t_0 b}(e^{EIP_D b} - 1)},
\tag{14}
$$

$$\mathbb{P}(z=j|\text{infected at age } t_0) =$$
$$\begin{cases} \mathbb{P}(z=j|\text{completes EIP at age } t_1)e^{-\frac{a}{b}e^{t_0 b}(e^{(t_1-t_0)b}-1)} \text{ if } j\neq 0 \\ \mathbb{P}(z=j|\text{completes EIP at age } t_1)e^{-\frac{a}{b}e^{t_0 b}(e^{t_1 b}-1)} + \left(1 - e^{-\frac{a}{b}e^{t_0 b}(e^{(t_1-t_0)b}-1)}\right) \text{ if } j=0 \end{cases}.$$
$$(15)$$

$\mathbb{E}(z|\text{infected at age } t_0)$ is an estimate of the lifetime number of infectious bites per mosquito (what we call rVC). To quantify temperature-dependent changes in transmission risk, we calculate how this quantity changes given the temperature-dependent changes in the constituent parameters estimated in the laboratory. We do not have measures of vector-host densities but assume the vector-host ratios and per-mosquito biting rates remain the same within a location. The predicted future mean temperature for one location, Garissa (30.7 °C), is marginally above our experimental range, so predicted future mean temperature for this location we used the upper temperature limit for which we have empirical data (30 °C). Analytical solutions in the integrands in Eqs. 9, 10, 11 and 15 were not available, so these were integrated numerically using the Cuhre method with an absolute tolerance of $10^{-7}$ (calculus R package[78]).

Since the temperatures we used in our studies do not align exactly with the estimates of temperature in the field settings, to parameterise mosquito survival when calculating the $\mathbb{E}(z|\text{infected at age } t_0)$, we fit temperature-dependent functions to the survival model parameter estimates, which we fit to data from each temperature independently. As mosquito survival differed between experiments, we estimate transmission potential separately for Feed1 and Feed2. To estimate the number of mosquitoes alive, we interpolate the Gompertz survival distributions to estimate daily cumulative survival rate at different temperatures. Values of parameter $a$ (initial mortality) of the Gompertz functions were fitted to a quadratic function,

$$G_q(T) = b_0 + b_1 T + b_2 T^2 \qquad (16)$$

where $G_q(T)$ is parameter $a$ of Gompertz function at a given temperature, $T$. Values of parameter $b$ (age-dependent mortality rate) fitted to a linear function,

$$G_l(T) = y + sT \qquad (17)$$

where $G_l(T)$ is parameter $b$ of Gompertz function at a given temperature, $T$. From these modelled quadratic and linear functions, Gompertz survival distributions were estimated for each temperature in the four Kenyan locations under current and future climate change scenarios.

For daily biting rate, we used individual mosquito gonotrophic cycle length data from a previously published study conducted on *A. stephensi*[24]. The temperature-dependence of the biting rate was estimated by fitting a Brière function,

$$B(T) = cT(T - T_0)(T_m - T)^{\frac{1}{2}}, \qquad (18)$$

to these data, where B(T) is biting rate at given temperature, $T$, $T_O$ and $T_m$ are the lower and upper threshold temperatures of biting rate, respectively. To do so, we assumed complete gonotrophic concordance and calculated the predicted gonotrophic cycle length as the inverse of the predicted per mosquito biting rate. We fitted the Brière function parameters in a Bayesian framework, assuming that the gonotrophic cycle lengths were geometrically distributed with the mean 1/B(T), using the Stan MCMC sampler (four chains each with 2500 iterations inclusive of 1250 warmup iterations; convergence was determined by $\hat{R} < 1.01$ for all parameters and by visually checking the trace plots). For the biting rate parameters, we assume weakly informative normal priors with the means equivalent to the values

estimated by Mordecai et al.[9] ($c$ ~ normal (0.000203, 0.01), $T_0$ ~ normal (11.7, 4.5) and $T_m$ ~ normal (42.3, 4.5)). The Brière function forces the biting rate to decline at temperature extremes, which is uncertain as the empirical data show no marked decline over the temperature range used. We note that *A. stephensi* is a different species, but this earlier research was conducted in the same insectary and rearing conditions as the current study, and biting rate is applied equivalently to the different models in any case.

## Statistical analysis
Mosquito survival was analysed using Kaplan–Meier Log Rank test to determine median survival and the effect of temperature on the survival (JMP 8.0, SAS). The mosquitoes collected for dissection were censored in the survival analysis. Least square estimation was used (Prism 8.02, GraphPad) for determining best fit survival function on survival data from Feed1 and Feed2 and for modelling temperature dependency of parameter values of the Gompertz survival functions.

## Reporting summary
Further information on research design is available in the Nature Portfolio Reporting Summary linked to this article.

## Data availability
All relevant data are available from the manuscript or https://github.com/IsaacStopard/mSOS_gambiae_only (https://doi.org/10.5281/zenodo.10688356). Source data are provided with this paper.

## Code availability
Code for the modelling study is available from https://github.com/IsaacStopard/mSOS_gambiae_only (https://doi.org/10.5281/zenodo.10688356).

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

## Acknowledgements

We thank Deonna C. Soergel and Janet L. Teeple for technical assistance. This study was supported by NIH NIAID grant # R01AI110793 and National Science Foundation Ecology and Evolution of Infectious Diseases grant (DEB-1518681) awarded by M.B.T.; I.J.S. was supported by PhD funding from the Natural Environment Research Council (NE/P012345/1) administered through the Quantitative Methods in Ecology and Evolution Centre for Doctoral Training. I.J.S. & T.S.C. received support from Wellcome Trust (226072/Z/22/Z) and the MRC Centre for Global Infectious Disease Analysis (reference MR/X020258/1), funded by the UK Medical Research Council (MRC). This UK funded award is carried out in the frame of the Global Health EDCTP3 Joint Undertaking; and acknowledges funding by Community Jameel. The funders had no role in study design, data collection and analysis, decision to publish, or preparation of the manuscript.

## Author contributions

E.S., I.J.S., B.L., J.L.W., T.S.C. and M.B.T. designed the research; E.S., J.L.W. and N.L.D. conducted the experiment; E.S. and I.J.S. analysed the data; and E.S., I.J.S., B.L., T.S.C. and M.B.T. wrote the manuscript with inputs from J.L.W. and N.L.D.

## Competing interests

The authors declare no competing interests.
