## [Peer Review File · Nature Communications]

Estimating of the effects of temperature on transmission of the human malaria parasite, *Plasmodium falciparum*REVIEWER COMMENTS

Reviewer #1 (Remarks to the Author):

This study used a mechanistic model which was fitted to experimental vector data to estimate the how the extrinsic incubation period, the mosquito to human transmission period, the adult survival rate and the vectoral capacity of anopheles gambiae for plasmodium falciparum malaria varied with temperature.

The EIP is a key parameter and portion of the malaria transmission cycle and how it changes with temperature has key implications for how vectoral capacity or R0 may vary with climate change.

Whilst others have explored this (Paaijmans et al 2012, doi:

<https://doi.org/10.1098/rsbl.2011.1075>), this work combines more complex models with detailed experimental data. This work will be of interest for those working on the impact of climate change on malaria vector biology.

Comments:

1. It was not fully clear to me from the methods, but it seems that the mosquitoes were reared within 27-30 degrees celsius, but only exposed to different temperatures after blood feeding. The parameters investigated are all observed following blood meals, but, given that studies of larval environmental conditions in mosquito vector species have found temperature and ecological factors such as competition impact important parameters related to vector competence (e.g. Westbrook et al., 2010 (doi: 10.1089/vbz.2009.0035), Alto et al., 2007 (doi: 10.1098/rspb.2007.1497) and Okech et al, 2007 (DOI: 10.1186/1475-2875-6-50)), larval environmental temperature may be an important consideration. If readers or the authors are looking to apply these parameter estimates to field settings or realistic climate change scenarios, which I assume they do given estimates of vectoral capacity are provided for specific sites, we would assume that mosquitos would be exposed to higher/lower environmental temperatures during their entire life history. I don't think this is a fatal flaw and other studies exploring temperature impacts on mosquito life history have reared mosquitoes in similar conditions, but this should be discussed. Particularly given that in the abstract the authors say that the paper " describe(s) how Plasmodium falciparum infection of the African malaria vector, Anopheles gambiae, is modulated by temperature, including its influences on parasite establishment, conversion efficiency through parasite developmental stages, EIP and overall competence".

2. Diurnal temperature fluctuations have been found to impact mosquito lifecycle and malaria transmission parameters, therefore the limitations of using a constant temperature should also be discussed. (Paaijmans et al 2010 , doi:10.1073/pnas.1006422107)

3. No measures of uncertainty or confidence are given for Relative vectorial capacity (rVC) - can the authors explain why uncertainty was not propagated through to this estimate? In addition, there could be more explanation of how vectoral capacity translates into malaria transmission metrics such as R0.

4. Generally methods are described well, however choice of priors for the MCMC could be more clearly described. I would recommend editing supplementary table 3 with an additional column giving the source/justification for each prior with references where appropriate.

5. A diagram or table illustrating the key metrics measured/modelled here and where they come into the malaria transmission cycle would be helpful.

6. Line 225 - why are the results given as fractions?

Reviewer #3 (Remarks to the Author):

This is a timely work that is needed and will play a significant role in the field of malaria and mathematical modeling or malaria transmission,

Noteworthy: The marked difference in the EIP estimates between this data-driven study and the established and commonly used degree-day model estimates, at constant temperatures outside the 24-28 degrees Celsius measurements, is noteworthy. In particular, at lower temperatures, for example 17 degrees Celsius, the EIP₅₀ (time for 50% of the mosquitoes to become infectious) estimates based on this study were much lower, 59-days, compared to the 111 days predicted estimates from the degree-day model. This obviously can be consequential for transmission estimates and preparedness at lower temperatures.

Also noteworthy is their parametrization of the efficiency of the conversion from oocyst to sporozoite infection at different constant temperatures. The indicated drop in efficiency at higher temperatures (28 & 30 degrees Celsius), as highlighted by their work, and 100% efficiency at most other temperatures, again highlights the novelty and originality in their work.

The paper is well written. The methodology is sound and well explained and I believe can be reproduced under similar conditions.

Reviewer #4 (Remarks to the Author):

This paper presents a comprehensive analysis of the role of temperature on the extrinsic incubation period of *Plasmodium falciparum* – first collecting experimental data on EIPs in *Anopheles gambiae*, before fitting a sophisticated mathematical model to the data to assess the role of temperature on a variety of salient quantities and extend a simplistic but established model of the temperature-EIP relationship. It is, in my view, a valuable addition to the literature.

The paper goes on to extrapolate their findings to assess the impact of potential future changes in climate to malaria transmission in the Kenyan highlands.

Experimental infection

I am not a laboratory scientist and therefore not qualified to comment on the experimental aspects of this study, although they appear consistent with other studies I have read.

Modelling

The authors proceed to use the data obtained from their infection experiments to model temperature-dependence of the extrinsic incubation period. The EIP is a vitally important parameter for understanding malaria transmission and understanding the effects of temperature on its distribution is critical to assess impact of climatic changes on malaria transmission.

The finding that the variance of the EIP distribution scales with the mean is interesting, given the lack of nuance around this in the standard degree-day model.

I would add some more commentary to Figure 2 in the text, pointing out the structural differences between this framework and the degree-day model.

Also re Figure 2 – the fit at 17C is very poor, but in such a way which suggests the degree-day model is even poorer. Why are the models performing so poorly there? This warrants some discussion and analysis, especially as you proceed to make inferences on a site with a mean temperature of 17.5C.

Kenya experiments

The study finds that in the coldest setting analysed the relative vectorial capacity is much (13 and 26x) higher than the established (but flawed) degree-day model. Under the examined future

climate scenario this setting also produces the largest divergence from the degree-day model, suggesting that the degree-day model vastly overestimates increase in transmission potential in this setting (but not as profoundly in the other three studied sites).

The large difference between the EIP models in Kericho is not well-reasoned, although there is some speculation in the discussion.

I think it needs to be made clearer in the text that the change in perspective here is that the degree-day model is not performing sufficiently well **currently**, and thus what has shifted (most) is our contemporary baseline understanding of EIP. This paper makes it clear (although it could be more explicit about it in the discussion) we need to better understand how EIP varies at the fringes of transmission.

Response letter to reviewers

Reviewer #1 (Remarks to the Author):

1. It was not fully clear to me from the methods, but it seems that the mosquitoes were reared within 27-30 degrees celsius, but only exposed to different temperatures after blood feeding. The parameters investigated are all observed following blood meals, but, given that studies of larval environmental conditions in mosquito vector species have found temperature and ecological factors such as competition impact important parameters related to vector competence (e.g. Westbrook et al., 2010 (doi: 10.1089/vbz.2009.0035), Alto et al., 2007 (doi: 10.1098/rspb.2007.1497) and Okech et al, 2007 (DOI: 10.1186/1475-2875-6-50)), larval environmental temperature may be an important consideration. If readers or the authors are looking to apply these parameter estimates to field settings or realistic climate change scenarios, which I assume they do given estimates of vectorial capacity are provided for specific sites, we would assume that mosquitos would be exposed to higher/lower environmental temperatures during their entire life history. I don't think this is a fatal flaw and other studies exploring temperature impacts on mosquito life history have reared mosquitoes in similar conditions, but this should be discussed. Particularly given that in the abstract the authors say that the paper “ describe(s) how Plasmodium falciparum infection of the African malaria vector, Anopheles gambiae, is modulated by temperature, including its influences on parasite establishment, conversion efficiency through parasite developmental stages, EIP and overall competence”.

Response: Thanks for the suggestion, we agree this is an important point. We have now clarified in the methods section that all developmental stages of mosquitoes were reared at $27^{\circ}\text{C}\pm 0.5^{\circ}\text{C}$, $80\%\pm 5\%$ relative humidity, and a 12h:12h light-dark photoperiod for all experiments.

We agree that environmental conditions for the larval stages may also influence EIP/vector competence and other life-history traits that shape vectorial capacity. We've added discussion on this point in line 367-375.

Also, just to be clear, our comparative modeling uses relative VC where we assume 'all else is equal', so any effects that might occur via larval conditions are equivalent between models.

2. Diurnal temperature fluctuations have been found to impact mosquito lifecycle and malaria

transmission parameters, therefore the limitations of using a constant temperature should also be discussed. (Paaijmans et al 2010 , doi:10.1073/pnas.1006422107)

Response: We agree this is an important point. The original data used to parameterize the Detinova model were based on constant temperature experiments so the use of constant temperatures in our study enables direct comparison. To address this limitation though we have already discussed the possible additional influence of temperature fluctuation (including the suggested reference) as one of the limitations of our study in the final paragraph (line 383).

3. No measures of uncertainty or confidence are given for Relative vectorial capacity (rVC) - can the authors explain why uncertainty was not propagated through to this estimate? In addition, there could be more explanation of how vectorial capacity translates into malaria transmission metrics such as R0.

Response: Thank you, the uncertainty for rVC has now been added. We propagated the uncertainty in the rVC estimates by calculating the rVC values (expected number of infectious bites per infected mosquito) using MCMC samples from our EIP and biting rate posterior estimates. Please see the updates to Table 1 and the Supplementary Fig. 5. All values have quoted in the text have also been updated.

We have also added additional text to explain rVC in relation to R0 (lines 206-216).

4. Generally methods are described well, however choice of priors for the MCMC could be more clearly described. I would recommend editing supplementary table 3 with an additional column giving the source/justification for each prior with references where appropriate.

Response: Thank you, this is a good point. We have now added details on the implications of our prior choices and suitable reference to justify these assumptions to Supplementary Table 6.

5. A diagram or table illustrating the key metrics measured/modelled here and where they come into the malaria transmission cycle would be helpful.

Response: We feel the equations and the related descriptions in the methods provide sufficient detail, but to add a source of extra information, we direct readers to a figure in an earlier publication¹ that provides a graphical summary of the approach.

6. Line 225 - why are the results given as fractions?

Response: This was to help explain how the fold change was calculated from the rVC values in Table 1. The 13-fold change was calculated by dividing 0.38 (Suh-Stopard) by 0.03 (Degree-day) for Feed1. Similarly, 26-fold was calculated by dividing 0.79 (Suh-Stopard) by 0.03 (Degree-day). Given we are presenting fold changes in a relative metric, we thought it useful to spell it out. We've added what these values mean in line 244.

Reviewer #3 (Remarks to the Author):

This is a timely work that is needed and will play a significant role in the field of malaria and mathematical modeling or malaria transmission,

¹ Stopard, I. J., Churcher, T. S. & Lambert, B. Estimating the extrinsic incubation period of malaria using a mechanistic model of sporogony. *PLoS Comput Biol* **17**, e1008658 (2021). <https://doi.org/10.1371/journal.pcbi.1008658>

Noteworthy: The mark difference in the EIP estimates between this data-driven study and the established and commonly used degree-day model estimates, at constant temperatures outside the 24-28 degrees Celsius measurements, is noteworthy. In particular, at lower temperatures, for example 17 degrees Celsius, the EIP₅₀ (time for 50% of the mosquitoes to become infectious) estimates based on this study were much lower, 59-days, compared to the 111 days predicted estimates from the degree-day model. This obviously can be consequential for transmission estimates and preparedness at lower temperatures.

Also noteworthy is their parametrization of the efficiency of the conversion from oocyst to sporozoite infection at different constant temperatures. The indicated drop in efficiency at higher temperatures (28 & 30 degrees Celsius), as highlighted by their work, and 100% efficiency at most other temperatures, again highlights the novelty and originality in their work.

The paper is well written. The methodology is sound and well explained and I believe can be reproduced under similar conditions.

Response: Thank you very much for spending the time to review our work; we appreciate the positive reviews on our manuscript.

Reviewer #4 (Remarks to the Author):

I would add some more commentary to Figure 2 in the text, pointing out the structural differences between this framework and the degree-day model.

Response: Our new EIP model enables us to present different measures (percentiles) of EIP to describe the temporal variation in sporogony in the mosquito population, while degree-day model assumes all mosquitoes have the same EIP. We presented representative EIPs such as EIP₁₀, EIP₅₀ and EIP₉₀ to describe this temporal variation. This has been clarified in line 149-152.

Also re Figure 2 – the fit at 17C is very poor, but in such a way which suggests the degree-day model is even poorer. Why are the models performing so poorly there? This warrants some discussion and analysis, especially as you proceed to make inferences on a site with a mean temperature of 17.5C.

Response: The main reason for poor fitting is that the infection at 17°C was extremely low as this temperature is very close to the lower thermal limit for vector competence, which adds a lot of variation to the temporal pattern of sporogony (i.e., from first infection maximum prevalence). However, even with high uncertainty, the EIP of degree-day model at 17°C is considerably larger than upper credible interval of our EIP₅₀, and accordingly the probability that our EIP₅₀ is less than EIP of degree-day model is close to 1 (see Fig. 2b). We have edited the text slightly to re-emphasize the poor model fit (line 274-275) and also the potential for non-linear effects at the thermal extremes, which could be a contributing factor (line 282-283).

Kenya experiments

The study finds that in the coldest setting analysed the relative vectorial capacity is much (13 and 26x) higher than the established (but flawed) degree-day model. Under the examined future climate scenario this setting also produces the largest divergence from the degree-day model, suggesting that the degree-day model vastly overestimates increase in transmission potential in this setting (but not as profoundly in the other three studied sites).

The large difference between the EIP models in Kericho is not well-reasoned, although there is some speculation in the discussion.

I think it needs to be made clearer in the text that the change in perspective here is that the degree-day model is not performing sufficiently well *currently*, and thus what has shifted (most) is our contemporary baseline understanding of EIP. This paper makes it clear (although it could be more explicit about it in the discussion) we need to better understand how EIP varies at the fringes of transmission.

Response: The reasons why our estimates of EIP are shorter than those of the Detinova model are unclear. We speculate about possible reasons (lines 276-293) but there's not too much we can add without further mechanistic research. However, in line with the reviewer's recommendation, we have made the discussion clearer that the differences in the impacts of climate change we predict between models for Kericho derive from the fact that the Detinova curve both underestimates current rates of parasite development and overestimates the rate of change in response to warming, relative to our model (lines 358-366).

REVIEWERS' COMMENTS

Reviewer #1 (Remarks to the Author):

The authors have addressed my comments and I am happy to recommend for publication